# Attention Deficit Hyperactivity Disorder Misdiagnosis: Why Medical Evaluation Should Be a Part of ADHD Assessment

**DOI:** 10.3390/brainsci13111522

**Published:** 2023-10-28

**Authors:** Joseph Sadek

**Affiliations:** Department of Psychiatry, Dalhousie University, Halifax, NS B3H4K3, Canada; joseph.sadek@nshealth.ca

**Keywords:** ADHD, diagnosis, medical evaluation, diabetes, thyroid dysfunction

## Abstract

Introduction: Attention deficit hyperactivity disorder (ADHD) is a neurodevelopmental disorder that interferes with multiple aspects of daily functioning and is associated with impairments in several domains. It may affect academic, educational, vocational, social, emotional, interpersonal, and health domains, and worsen risks to health outcomes. Objective: To identify and discuss medical conditions that commonly present with symptoms resembling ADHD. Method: This review is selective and not systematic. It is conducted through a focused literature search through PubMed, Google Scholar, and EMBASE. Search term included “ADHD misdiagnosis”, “medical conditions with ADHD like symptoms”, “ADHD AND medical problems”. Exclusion: giftedness, high IQ, and any article that does not list medical conditions. The limits applied were the following: the work must have been published in the past 20 years, be on humans, and be in the English language. Results: There are several medical conditions that can be misdiagnosed as ADHD and may show a similar presentation to ADHD, particularly with inattentive symptoms. Examples include, but are not limited to, absence seizure disorder, diabetes, thyroid dysfunction, sleep deprivation, post-concussion states, inflammatory bowel disease, iron deficiency states and anemia, and disordered breathing. Conclusions: Our review suggests that a thorough medical evaluation should be conducted prior to the diagnosis of ADHD. Allied health professionals and psychologists who diagnose ADHD should seek medical clearance from a physician prior to making the ADHD diagnosis in order to reduce misdiagnosis rates and improve patient outcomes. ADHD diagnosis should follow guidelines and be carried out under a systematic standardized approach. A full medical evaluation should be conducted to assess for medical conditions that may look like ADHD or be associated with ADHD.

## 1. Introduction

Attention deficit hyperactivity disorder (ADHD) is a neurodevelopmental disorder that interferes with multiple aspects of daily functioning. The diagnosis requires the presence of developmentally inappropriate levels of inattentive and/or hyperactive–impulsive symptoms, at least six symptoms in children and adolescents and five in adults, lasting for at least 6 months, occurring in different settings, and first manifesting in childhood prior to the age of 12 [1,2]. The presentation of the disorder is characterized by inattention, hyperactivity, or both [3]. Symptoms of inattention often include difficulty in paying attention to details, being easily distracted, avoiding tasks requiring sustained attention, not listening while spoken to directly, difficulty sustaining attention in tasks, disorganization, starting tasks but not finishing them, and not remembering things such as appointments or paying bills. Symptoms of hyperactivity–impulsivity include often running or climbing rather than walking, talking excessively, being loud, difficulty waiting one’s turn, interrupting others, answering questions before they are complete, going non-stop all day, difficulty with prolonged seating, and fidgeting [1,2]. The reported prevalence of the disorder ranges from 3% to 8% in school-aged children and from 1.5% to 3% in adults. The wide variation in prevalence results from several factors, including the diagnostic criteria used, the age and gender of study populations, and geographic areas [4,5,6,7]. The prevalence of ADHD decreases with age, and the disorder persists into adulthood in approximately 40% of affected children [4].

ADHD is associated with impairments in several domains. It may affect academic, educational, vocational, social, emotional, interpersonal, and health domains, and worsen risks to health outcomes [8,9].

Diagnosis of many psychiatric disorders cannot be made if the disorder is attributable to physiological effects of substance or another medical condition. DSM-5 is the diagnostic and statistical manual that is used by clinicians for the diagnosis of mental disorders; however, the DSM-5 diagnostic criteria for ADHD do not contain the exclusion phrase that applies to several other disorders such as depression [1,2]. ADHD diagnosis is challenging, particularly in the adult population. The current diagnosis of ADHD is based on the number of symptoms, presence of symptoms in two settings, presence of some symptoms prior to age 12, and functional impairment. Several researchers suggested that ADHD symptoms could be present secondary to a general medical condition or substance without the diagnosis of ADHD [10].

The objective of this review is to identify and discuss medical conditions that commonly present with symptoms resembling ADHD.

## 2. Materials and Methods

This review is selective rather than systematic and comprehensive. It attempts to provide guidance on several questions that are relevant to clinicians, such as what general medical conditions could produce symptoms like these of ADHD. A focused literature search was conducted in PubMed, Google Scholar, and EMBASE, using the search terms: “Attention deficit hyperactivity disorder”, “ADHD”, “attention”, “hyperactivity”, “impulsivity”, and “general medical condition”, “ADHD”, and any of the following: “endocrine”, “intellectual disability”, “sleep”, “seizure”, and “anemia”), “ADHD misdiagnosis”, “medical conditions with ADHD like symptoms”, “ADHD and medical problems”. Exclusion: ADHD and giftedness, high IQ. Any article that does not list medical conditions was excluded. The relevant articles were selected and reviewed by the primary investigator. Few articles linked ADHD to medical conditions, but several articles described inattentive symptoms as part of the presenting symptoms of the medical condition.

The limits applied were the following: works must published in the past 20 years, be about humans, and be in the English language. The relevant articles were reviewed, and the important findings are summarized below.

## 3. Results

### 3.1. Seizure Disorder: Absence Seizure

Absence seizures can be typical or atypical. Typical absence seizures (TASs) are brief (i.e., 4–20 s) and frequent, with an abrupt and severe loss of consciousness, characterized by gaze rigidity, freezing, and sometimes automatisms and chewing. The eyes may turn upwards and eyelids flutter. It usually lasts less than 10 s. They are brief, so they can be easily missed. The ictal electroencephalogram shows generalized discharges of high-amplitude spikes and waves around 3 Hz. Seizures appear during wakefulness in children with normal neurological examination and development. Atypical seizures last longer and may extend beyond 20 s. They have a slower onset and offset with a change in muscle tone and movement [11].

The clinical presentation of children with TASs could be misdiagnosed as the inattentive subtype of ADHD (e.g., daydreaming, and appearing glazed, spaced out, blank, clumsy, and unable to sustain attention). The inattentive ADHD-like presentation may depend on epilepsy-related characteristics such as the seizure frequency, duration of the illness, and antiepileptic medications [12]. Missing the diagnosis of seizure disorder may have serious implications. Prescribing stimulants to patients with seizure may lead to increased seizure activity and a very poor quality of life. More serious implications may happen if a patient loses consciousness resulting in serious head trauma. Patients with seizure disorder require antiepileptic treatment, not psychostimulants.

Patients with TASs can also have comorbid anxiety, linguistic, and sleep disorders [13,14].

ADHD can also be misdiagnosed as absence seizures. Children with ADHD often present sleep problems Sleep disorders could lead to a behavioral and/or emotional dysregulation characterized by hyperactivity and impulsivity. Also, vigilance level disturbances can mimic absence seizures [15,16,17].

### 3.2. Diabetes Mellitus

Diabetes comprises a group of metabolic syndromes characterized by hyperglycemia. Type 1 diabetes (T1DM) is called insulin-dependent, and type 2 is called noninsulin-dependent DM. Disturbances to the blood glucose can lead to a series of pathogenetic mechanisms involving neurological dysfunction, including neuronal apoptosis, abnormal energy metabolism, synaptic dysfunction, neurodegenerative changes, and oxidative stress in brain tissue [18,19,20,21].

In T1DM, cognitive problems have been studied in several age groups. Some studies have suggested an impaired executive function and poor memory between the ages of 9 and 19. A lower verbal intelligence was noticed in the young T1DM group than in the non-diabetic control group [22,23,24].

Lower academic achievements were reported in students with T1DM than their classmates who are non-diabetic [25].

Adult patients (>22 years old) with T1DM also showed cognitive changes in several studies. They showed a reduced sustained attention [26] and information-processing speed [27]. They also showed impaired executive functions in concept formation, anticipation, and cognitive flexibility [28]. An earlier age of onset of T1DM was associated with a worse cognitive performance.

Hyperglycemia or hypoglycemia can also be found in T1DM due to difficulty regulating glucose metabolism. Chronic hyperglycemia is associated with cognitive impairment, visuospatial abilities, and information processing, particularly with an early age of onset and long duration of illness [29,30].

Diabetic ketoacidosis is a complication of T1DM, and pediatric patients newly diagnosed with both show a clear pattern of poorer cognitive functioning, and they perform poorly in executive functioning [31].

Cognitive impairment in type 2 diabetes (T2DM) may be related to reduced activation in the hemisphere temporoparietal regions [32]. Damage to some areas in the temporal and parietal lobes of the brain may lead to memory deficits, a deficit in executive functions, and inattention [33].

There are additional metabolic problems on top of glucose control, such as disorders of lipid metabolism and increased levels of ghrelin, cortisol, and C-reactive protein, that may also lead to a decrease in neurological function [34,35]. Research at the gene level may also help to explain the association between ADHD and diabetes.

In patients with T2DM, the single-nucleotide polymorphisms rs17518584 and Hp1-1 in patients with T2DM are associated with executive functions and attention/working memory, respectively [36,37].

### 3.3. Thyroid Dysfunction

#### 3.3.1. Subclinical Hypothyroidism (SCH)

SCH cannot be diagnosed via clinical findings, but via an elevated serum TSH concentration and normal circulating free T_4_ and T_3_ concentrations. Subclinical hyperthyroidism has been associated with cognitive decline in some but not all studies [38].

Younger people with SCH were observed to have mild cognitive impairment with subtle deficits in memory, executive function, difficulties in new learning, and attention problems [39]. Several studies suggested that major alterations in cognitive function do not reliably improve with levothyroxine therapy [40].

#### 3.3.2. Hypothyroidism

Hypothyroidism can affect several cognitive domains [41,42,43,44] and particularly a deficit in verbal memory [44,45]. Several studies have reported deficits in general intelligence, attention or concentration, visuospatial processing, working memory, motor speed, perceptual function, language, psychomotor function, and executive function. Levothyroxine treatment is usually effective in treating these decrements, although a complete reversal may not be achieved [42,46,47,48,49].

#### 3.3.3. Hashimoto Thyroiditis

Hashimoto thyroiditis (HT) is the most common autoimmune disease of the thyroid gland that is characterized by the infiltration of hematopoietic mononuclear cells, mainly lymphocytes, in the thyroid follicles. The term HT includes the classic form, and other forms that are less prevalent, such as the fibrous variant, IgG4-related variant, juvenile form (from the age of 10 to 18), Hashitoxicosis, and painless thyroiditis (sporadic or post-partum). All forms are characterized pathologically by thyroid cells undergoing atrophy or transforming into a bolder type of follicular cell, rich in mitochondria, called a Hürthle cell (HT). Most HT forms ultimately evolve into hypothyroidism, although, at presentation, patients can be euthyroid or even hyperthyroid [50,51].

The diagnosis of HT relies on the presence of circulating antibodies to thyroid antigens (mainly thyroperoxidase and thyroglobulin) and reduced echogenicity on a thyroid ultrasound [52].

There are several cognitive clinical manifestations of HT and hypothyroidism. An inability to concentrate, difficulty focusing, executive dysfunction, and poor memory are common and are reported by the majority of patients. It can be confused with ADHD, particularly early in the disease, when the patient is euthyroid. Local manifestations due to compression of the cervical structures that are anatomically close to the thyroid gland may include dysphonia, dyspnea, and dysphagia. Systemic manifestations originate from a loss of function of the thyroid gland and subsequent primary hypothyroidism. Examples of systemic manifestations include constipation, decreased peristalsis, and dry, cold, yellowish, and thickened skin, in addition to several cardiovascular implications. Muscles appear falsely hypertrophic. The respiratory system, hematopoietic system, and reproductive system may also be affected. Depression is also common. Diagnosis can be made through ultrasound, which has replaced fine-needle biopsy [53,54,55,56,57].

### 3.4. Sleep Deprivation

Sustained or vigilant attention means the ability to maintain stable, focused attention across a time interval [58]. It is a major component of a wide range of cognitive performance tasks. Sleep deprivation can result in impairment in vigilant attention.

Consecutive days of sleep restriction may lead to a cumulative deficit in sustained attention [59].

Patients with sleep deprivation can be misdiagnosed as having ADHD due to difficulty in sustaining attention. It is important to inquire about sleep quantity and quality in patients assessed for ADHD. A recent study suggested that sleep deprivation in early childhood is associated with a higher risk of ADHD in middle childhood [60].

Poor sleep could lower inhibitory control and increase impulsivity [61]. There may be an overlap between neurocognitive deficits in children with ADHD and those affecting healthy people with a poor sleep quality [62].

Sleep alterations may produce ADHD-like symptoms and, on the other hand, an altered sleep architecture could exacerbate ADHD symptoms [16].

The association of sleep with ADHD is quite complex. Problems with sleep may be an intrinsic feature of ADHD. Regular sleep disturbance can lead to the development of ADHD or ADHD-like symptoms, potentially resulting in misdiagnosis [63,64].

The effects of restricted, disordered, or disrupted sleep can manifest as symptoms, behaviors, or functional impairments that are remarkably like those of ADHD [65,66]. The interrelationships are further complicated through the use of psychostimulant medications to treat ADHD, which impair sleep in some patients [67] but paradoxically improve sleep in others via a calming effect [68]. For these reasons, it has been recommended that primary sleep disorders should be ruled out before initiating ADHD medication [63]. Behavioral interventions targeted at improving sleep may benefit some patients [69] and should form part of the multimodal ADHD management plan recommended for patients receiving pharmacotherapy [70].

Children with inadequate sleep may present with poor attention and behavioral symptoms that overlap with the core features of ADHD [62]. Some studies have suggested that a short sleep duration correlates with ADHD-like symptoms and behaviors scored by parents [71,72] and teachers [73].

### 3.5. Sleep-Disordered Breathing and Obstructive Sleep Apnea

Sleep-disordered breathing (SDB) includes several conditions such as obstructive sleep apnea (OSA) and primary snoring [74]. SDB has been consistently associated with neurobehavioral and neurocognitive deficits, including inattentive or ADHD-like symptoms [75,76,77,78]. Other research has concluded that the prevalence of OSA in patients with ADHD (25–30%) is higher than in the general population (about 3%) [79], and some guidelines have suggested that patients undergoing evaluation for ADHD should be assessed for sleep apnea [70]. There are questionnaires, portable devices, and polysomnography that can be used to diagnose OSA.

### 3.6. Other Conditions

Post-concussion states, inflammatory bowel disease, iron-deficiency states, and anemia can present with symptoms of inattention that can be confused with the inattentive type of ADHD.

## 4. Discussion and Conclusions

The present review supports the notion that ADHD symptoms could be present secondary to a general medical condition, without the diagnosis of ADHD. Since the DSM-5 diagnostic criteria for ADHD do not contain an exclusion phrase that applies to several other disorders such as depression, we suggest a change in the DSM-6 to exclude the diagnosis of ADHD if the ADHD symptoms are attributable to physiological effects of substance or another medical condition.

The review also suggests that a thorough medical evaluation should be conducted prior to the diagnosis of ADHD to reduce misdiagnosis rates and improve patient outcomes. Allied health professionals and psychologists who diagnose ADHD should seek medical clearance from a physician prior to making the ADHD diagnosis. ADHD diagnosis should follow guidelines and be carried out in a systematic, standardized way. A full medical evaluation should be conducted to assess medical conditions that may look like ADHD or be associated with ADHD.

A misdiagnosis of ADHD and missing an important medical diagnosis such as seizure disorder, diabetes in children, or thyroid dysfunction may have life-threatening consequences and long-term implications on the quality of life of patients.

Since hypothyroidism may present with cognitive difficulties, serum TSH should be ordered in patients with impaired cognitive function. Cognitive deficits related to thyroid dysfunction are largely reversible with levothyroxine therapy.

The majority of the conditions listed in this review can be easily ruled out by a qualified medical practitioner and, at times, a simple blood test. It is understood that the general medical conditions that produce ADHD-like symptoms are rare and, in the majority of cases, there are clear signs to suggest the presence of a medical condition. Therefore, considering the secondary causes of ADHD can optimize patient care and should be the standard of care.

Prior to an ADHD diagnosis, routine blood work should be ordered, and additional tests should include thyroid stimulating hormone (TSH), Hemoglobin A1c, complete blood count (CBC), an obstructive sleep apnea (OSA) questionnaire, and taking the patient history regarding seizures and sleep. These may help to rule out general medical conditions that produce ADHD-like symptoms. Any family history of sudden death and cardiovascular problems should also be obtained.

Medical evaluation is also important as ADHD diagnosis can be missed in some medical conditions, such as chromosome 15 microdeletion syndromes or Fragile X.

Clinicians who are faced with pressure to diagnose ADHD should seek support from colleagues and professional organizations to ensure that societal pressure does not lead to a misdiagnosis of ADHD. Patients might find the delay in ADHD diagnosis to order medical investigations frustrating, but when the rational of the delay is explained, most patients may accept that the process aims to protect them and ensure that they receive the appropriate diagnosis.

ADHD diagnosis remains challenging and complex. Lauria’s neuropsychological theory adds another dimension of complexity to brain functioning, particularly due to the complex representation of the cultural psychological process in the central nervous system. It was emphasized that neuropsychological assessment helps in understanding the complex functional changes in the human brain, rather than localizing a lesion. Further research is needed in the diagnostic process of ADHD in adults.

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
