# Peer review of "Attention Deficit Hyperactivity Disorder Misdiagnosis: Why Medical Evaluation Should Be a Part of ADHD Assessment"

_brainsci, 2023, doi:10.3390/brainsci13111522_

Round 1
Reviewer 1 Report
Comments and Suggestions for Authors
Dear authors, I agree that it is necessary to provide medical assessment prior to diagnosis of ADHD. But the most important point in the current critic of this label is the presence of uncertain list of very general symptoms. I would say that ADHD is an umbrella, which put together any cases and any kind of combinations of clinical pictures and medical conditions. The serious problems is that DSM-5 doesn't care at all about any kind of medical condition. The only interest of ADHD is to state that the person presents ADHD and never to find any kind of the reason or concern of real situation of the patient. The critic is really necessary today. But, it is necessary to include the description of qualitative neuropsychological assessment, and it is provided by representatives of cultural historical neuropsychology. Please, look for publications by Glozman, Solovieva, Machinskaya and others followers of Luria's conception.
Author Response
Dear Reviewer
I am extremely grateful for the time, energy, and expertise that you devoted for this review. I agree with your comment that the serious problem is that DSM-5 doesn't care at all about any kind of medical condition. Unfortunately, I have seen children misdiagnosed with ADHD and they had type I diabetes that was not diagnosed. Once they started Insulin, their inattentive symptoms much improved.
I agree that qualitative neuropsychological assessment description is important, but I will elaborate on it in the second part of this publication. I am in process of writing a systematic review on different neuropsychological assessment and different rating scales used in ADHD assessment.
The book and publications by Glozman on the contributions of Luria and Reitan to developmental neuropsychology is extremely helpful and will be cited in my next review.
Thanks again for the excellent review.
Joseph Sadek

Reviewer 2 Report
Comments and Suggestions for Authors
Abstract
v Reorganize the abstract with clear section headings for Objective, Methods, Results, and Conclusion.
v The stated objective is vague and lacks specificity. It mentions gathering evidence of medical conditions with ADHD-like symptoms but does not clarify the primary research question or hypothesis.
v Refine the objective by specifying the exact research question or hypothesis, such as "To identify and discuss medical conditions that commonly present with symptoms resembling ADHD."
v The abstract mentions that the review is "selective and not systematic." This raises concerns about the rigor and methodology employed in the review. It does not provide information on inclusion/exclusion criteria, search strategy, or the number of studies reviewed.
v Include a brief description of the methodology, such as the search terms used, databases searched, and any inclusion/exclusion criteria applied.
v The abstract briefly mentions "several medical conditions" without specifying or quantifying them. Readers need more information on the identified conditions and their relevance to ADHD misdiagnosis.
v Suggestion for Improvement: Provide a concise summary of the specific medical conditions identified in the review and their shared symptoms with ADHD.
v The conclusion recommends a thorough medical evaluation before diagnosing ADHD and suggests involving physicians for medical clearance. While this is a valid point, the abstract does not mention the implications or potential impact of this recommendation.
v Suggestion for Improvement: Expand the conclusion to discuss the potential benefits of a thorough medical evaluation in reducing misdiagnosis rates and improving patient outcomes.
v The abstract could benefit from improved language clarity and conciseness. For example, the sentence, "ADHD diagnosis should be carried in a systematic way," could be phrased more clearly.
v Carefully revise the abstract to enhance readability and eliminate ambiguous language
Introduction
- The introduction provides a clear definition of ADHD, which is essential. However, it would be helpful to briefly mention the primary symptoms of ADHD (inattention, hyperactivity, impulsivity) for readers who may not be familiar with the disorder.
- Include a brief mention of the primary symptoms of ADHD to provide context.
- While prevalence data is mentioned, it would be beneficial to briefly discuss the impact of these prevalence rates and why they vary. For instance, you could mention that the variation in prevalence is due to factors such as diagnostic criteria and geographical differences.
- Provide a brief explanation of the factors contributing to the variation in ADHD prevalence rates.
- The introduction should smoothly transition from providing general information about ADHD to the specific topic of the review (medical conditions misdiagnosed as ADHD). Currently, this transition is abrupt.
- Use a transitional sentence or paragraph to link the general information about ADHD to the main topic of the review.
- The introduction mentions the DSM-5 criteria for ADHD but does not explain what DSM-5 stands for or its significance. Providing a brief explanation would be beneficial, especially for readers who may not be familiar with this terminology.
- Briefly explain what DSM-5 stands for and why it is relevant to the diagnosis of ADHD.
- While the objective is mentioned, it could be more explicitly stated. The objective should clearly state the purpose of the review and what readers can expect to gain from it.
- Reformulate the objective to explicitly state the goal of gathering evidence on medical conditions that mimic ADHD symptoms
- For enrichment of Introduction, please use the below references
https://link.springer.com/article/10.1007/s00787-023-02148-1
https://www.tandfonline.com/doi/abs/10.1080/15374416.2020.1867989
Materials and Methods
The statement that this review is "selective rather than systematic and comprehensive" is clear, but it's important to explicitly state the criteria used for selecting studies. Readers should understand why certain articles were chosen and others excluded.
Specify the criteria or rationale used for selecting the literature to ensure transparency.
The section mentions that the review aims to answer relevant questions for clinicians, such as identifying general medical conditions that could mimic ADHD symptoms. While this is mentioned, it would be beneficial to explicitly state the research questions or objectives guiding the review.
Clearly outline the research questions or objectives to provide a roadmap for the review.
The search terms used for PubMed and EMBASE are mentioned, which is good. However, it's advisable to provide more details about the search strategy, such as the Boolean operators used, any additional filters applied, and the date when the search was conducted.
Provide a more detailed description of the search strategy used, including specific terms, operators, and any filters.
The section briefly mentions limits applied, such as the time frame (past 20 years), language (English), and the human study population. It would be helpful to explain the rationale behind these criteria and whether any other criteria were used to select articles.
Provide a rationale for the inclusion and exclusion criteria and mention if any other criteria were applied.
It is mentioned that "relevant articles were reviewed," but it would be helpful to provide information on how the articles were reviewed. For example, mention if articles were screened independently by multiple reviewers and how discrepancies were resolved.
Describe the review process in more detail, including how articles were screened and any methods used to ensure consistency.
The section briefly mentions that "important findings are summarized below," but it doesn't provide any specific findings. Readers would benefit from a brief overview of key findings or themes identified during the review process.
Provide a concise summary of the important findings or themes discovered during the review.
Results
- The section is divided into subsections discussing different medical conditions, which is a good organizational approach. However, it would be helpful to have clear headings or titles for each subsection to guide readers through the content.
- Add clear and descriptive subsection headings for each medical condition (e.g., "3.1. Seizure Disorder: Absence Seizure," "3.2. Diabetes Mellitus," etc.).
- The descriptions of each medical condition are generally informative, but some sections could benefit from more detail. For instance, when discussing seizure disorders, you mention that absence seizures can be mistaken for ADHD but do not discuss the potential consequences or implications of such misdiagnoses in detail.
- Suggestion for Improvement: Provide additional details on the potential consequences of misdiagnosis for each medical condition discussed, including how it might affect treatment and patient outcomes.
- The section mentions that sleep deprivation can lead to symptoms resembling ADHD, but it would be beneficial to explicitly connect this information with the previous sections on medical conditions. For instance, how might sleep deprivation interact with or exacerbate symptoms in patients with these medical conditions?
- Establish clear connections between the different medical conditions and their potential overlap with sleep-related issues.
- Consider including a brief summary or concluding statement at the end of each subsection to highlight the key points and findings related to each medical condition.
- Conclude each subsection with a brief summary or key takeaway regarding the medical condition discussed.
- Ensure that there is a smooth flow and transition between subsections. Each subsection should naturally lead to the next without abrupt shifts in topic.
- Use transitional sentences or paragraphs to connect the different medical conditions and maintain a cohesive narrative.
- Consider including figures or tables to visually represent key findings or data related to each medical condition. Visual aids can enhance understanding.
- Create figures or tables, if applicable, to illustrate key points or data within each subsection.
Discussion and Conclusions
ü The suggestion to modify the DSM-6 criteria to exclude the diagnosis of ADHD when symptoms are attributable to medical conditions is a significant proposal. However, it would be beneficial to provide a more detailed rationale for this recommendation. Explain why it's important, how it would improve diagnosis, and any potential challenges or benefits of such a change.
ü Elaborate on the rationale and potential implications of the proposed change to the DSM criteria.
ü While you emphasize the importance of a thorough medical evaluation, it would be helpful to provide practical guidance on what this evaluation should entail. What specific medical tests or assessments should be part of this evaluation, and who should conduct it?
ü Offer specific recommendations for the components of a thorough medical evaluation.
ü The section briefly mentions that misdiagnosis can have life-threatening consequences and long-term implications. It would be valuable to provide concrete examples or case studies illustrating these consequences to underscore the significance of your findings.
ü Include specific examples or case studies to illustrate the potential consequences of misdiagnosis.
ü You mention that cognitive deficits related to thyroid dysfunction are largely reversible with levothyroxine therapy. It would be helpful to discuss the reversibility of cognitive deficits in the context of other medical conditions discussed in the review.
ü Extend the discussion on the potential reversibility of cognitive deficits in various medical conditions.
ü Consider addressing the accessibility of thorough medical evaluations, particularly for individuals who may have limited access to healthcare resources. Are there challenges or disparities in accessing these evaluations that should be acknowledged?
ü Discuss potential challenges or disparities in accessing thorough medical evaluations.
ü The section briefly mentions that ADHD diagnosis can be missed in some medical conditions like chromosome 15 microdeletion syndromes and Fragile X. Expanding on these conditions and their diagnostic challenges could be informative.
ü Provide more information on specific medical conditions where ADHD diagnosis may be overlooked.
Comments on the Quality of English LanguageAbstract
v Reorganize the abstract with clear section headings for Objective, Methods, Results, and Conclusion.
v The stated objective is vague and lacks specificity. It mentions gathering evidence of medical conditions with ADHD-like symptoms but does not clarify the primary research question or hypothesis.
v Refine the objective by specifying the exact research question or hypothesis, such as "To identify and discuss medical conditions that commonly present with symptoms resembling ADHD."
v The abstract mentions that the review is "selective and not systematic." This raises concerns about the rigor and methodology employed in the review. It does not provide information on inclusion/exclusion criteria, search strategy, or the number of studies reviewed.
v Include a brief description of the methodology, such as the search terms used, databases searched, and any inclusion/exclusion criteria applied.
v The abstract briefly mentions "several medical conditions" without specifying or quantifying them. Readers need more information on the identified conditions and their relevance to ADHD misdiagnosis.
v Suggestion for Improvement: Provide a concise summary of the specific medical conditions identified in the review and their shared symptoms with ADHD.
v The conclusion recommends a thorough medical evaluation before diagnosing ADHD and suggests involving physicians for medical clearance. While this is a valid point, the abstract does not mention the implications or potential impact of this recommendation.
v Suggestion for Improvement: Expand the conclusion to discuss the potential benefits of a thorough medical evaluation in reducing misdiagnosis rates and improving patient outcomes.
v The abstract could benefit from improved language clarity and conciseness. For example, the sentence, "ADHD diagnosis should be carried in a systematic way," could be phrased more clearly.
v Carefully revise the abstract to enhance readability and eliminate ambiguous language
Introduction
- The introduction provides a clear definition of ADHD, which is essential. However, it would be helpful to briefly mention the primary symptoms of ADHD (inattention, hyperactivity, impulsivity) for readers who may not be familiar with the disorder.
- Include a brief mention of the primary symptoms of ADHD to provide context.
- While prevalence data is mentioned, it would be beneficial to briefly discuss the impact of these prevalence rates and why they vary. For instance, you could mention that the variation in prevalence is due to factors such as diagnostic criteria and geographical differences.
- Provide a brief explanation of the factors contributing to the variation in ADHD prevalence rates.
- The introduction should smoothly transition from providing general information about ADHD to the specific topic of the review (medical conditions misdiagnosed as ADHD). Currently, this transition is abrupt.
- Use a transitional sentence or paragraph to link the general information about ADHD to the main topic of the review.
- The introduction mentions the DSM-5 criteria for ADHD but does not explain what DSM-5 stands for or its significance. Providing a brief explanation would be beneficial, especially for readers who may not be familiar with this terminology.
- Briefly explain what DSM-5 stands for and why it is relevant to the diagnosis of ADHD.
- While the objective is mentioned, it could be more explicitly stated. The objective should clearly state the purpose of the review and what readers can expect to gain from it.
- Reformulate the objective to explicitly state the goal of gathering evidence on medical conditions that mimic ADHD symptoms
- For enrichment of Introduction, please use the below references
https://link.springer.com/article/10.1007/s00787-023-02148-1
https://www.tandfonline.com/doi/abs/10.1080/15374416.2020.1867989
https://journals.sagepub.com/doi/abs/10.1177/10731911211003967
Materials and Methods
The statement that this review is "selective rather than systematic and comprehensive" is clear, but it's important to explicitly state the criteria used for selecting studies. Readers should understand why certain articles were chosen and others excluded.
Specify the criteria or rationale used for selecting the literature to ensure transparency.
The section mentions that the review aims to answer relevant questions for clinicians, such as identifying general medical conditions that could mimic ADHD symptoms. While this is mentioned, it would be beneficial to explicitly state the research questions or objectives guiding the review.
Clearly outline the research questions or objectives to provide a roadmap for the review.
The search terms used for PubMed and EMBASE are mentioned, which is good. However, it's advisable to provide more details about the search strategy, such as the Boolean operators used, any additional filters applied, and the date when the search was conducted.
Provide a more detailed description of the search strategy used, including specific terms, operators, and any filters.
The section briefly mentions limits applied, such as the time frame (past 20 years), language (English), and the human study population. It would be helpful to explain the rationale behind these criteria and whether any other criteria were used to select articles.
Provide a rationale for the inclusion and exclusion criteria and mention if any other criteria were applied.
It is mentioned that "relevant articles were reviewed," but it would be helpful to provide information on how the articles were reviewed. For example, mention if articles were screened independently by multiple reviewers and how discrepancies were resolved.
Describe the review process in more detail, including how articles were screened and any methods used to ensure consistency.
The section briefly mentions that "important findings are summarized below," but it doesn't provide any specific findings. Readers would benefit from a brief overview of key findings or themes identified during the review process.
Provide a concise summary of the important findings or themes discovered during the review.
Results
- The section is divided into subsections discussing different medical conditions, which is a good organizational approach. However, it would be helpful to have clear headings or titles for each subsection to guide readers through the content.
- Add clear and descriptive subsection headings for each medical condition (e.g., "3.1. Seizure Disorder: Absence Seizure," "3.2. Diabetes Mellitus," etc.).
- The descriptions of each medical condition are generally informative, but some sections could benefit from more detail. For instance, when discussing seizure disorders, you mention that absence seizures can be mistaken for ADHD but do not discuss the potential consequences or implications of such misdiagnoses in detail.
- Suggestion for Improvement: Provide additional details on the potential consequences of misdiagnosis for each medical condition discussed, including how it might affect treatment and patient outcomes.
- The section mentions that sleep deprivation can lead to symptoms resembling ADHD, but it would be beneficial to explicitly connect this information with the previous sections on medical conditions. For instance, how might sleep deprivation interact with or exacerbate symptoms in patients with these medical conditions?
- Establish clear connections between the different medical conditions and their potential overlap with sleep-related issues.
- Consider including a brief summary or concluding statement at the end of each subsection to highlight the key points and findings related to each medical condition.
- Conclude each subsection with a brief summary or key takeaway regarding the medical condition discussed.
- Ensure that there is a smooth flow and transition between subsections. Each subsection should naturally lead to the next without abrupt shifts in topic.
- Use transitional sentences or paragraphs to connect the different medical conditions and maintain a cohesive narrative.
- Consider including figures or tables to visually represent key findings or data related to each medical condition. Visual aids can enhance understanding.
- Create figures or tables, if applicable, to illustrate key points or data within each subsection.
Discussion and Conclusions
ü The suggestion to modify the DSM-6 criteria to exclude the diagnosis of ADHD when symptoms are attributable to medical conditions is a significant proposal. However, it would be beneficial to provide a more detailed rationale for this recommendation. Explain why it's important, how it would improve diagnosis, and any potential challenges or benefits of such a change.
ü Elaborate on the rationale and potential implications of the proposed change to the DSM criteria.
ü While you emphasize the importance of a thorough medical evaluation, it would be helpful to provide practical guidance on what this evaluation should entail. What specific medical tests or assessments should be part of this evaluation, and who should conduct it?
ü Offer specific recommendations for the components of a thorough medical evaluation.
ü The section briefly mentions that misdiagnosis can have life-threatening consequences and long-term implications. It would be valuable to provide concrete examples or case studies illustrating these consequences to underscore the significance of your findings.
ü Include specific examples or case studies to illustrate the potential consequences of misdiagnosis.
ü You mention that cognitive deficits related to thyroid dysfunction are largely reversible with levothyroxine therapy. It would be helpful to discuss the reversibility of cognitive deficits in the context of other medical conditions discussed in the review.
ü Extend the discussion on the potential reversibility of cognitive deficits in various medical conditions.
ü Consider addressing the accessibility of thorough medical evaluations, particularly for individuals who may have limited access to healthcare resources. Are there challenges or disparities in accessing these evaluations that should be acknowledged?
ü Discuss potential challenges or disparities in accessing thorough medical evaluations.
ü The section briefly mentions that ADHD diagnosis can be missed in some medical conditions like chromosome 15 microdeletion syndromes and Fragile X. Expanding on these conditions and their diagnostic challenges could be informative.
ü Provide more information on specific medical conditions where ADHD diagnosis may be overlooked.
Author Response
Dear Reviewer
I am extremely grateful for the excellent points that you raised in each section.
I made all the requested changes in each area in the attached manuscript.
-Objective has been changed "To identify and discuss medical conditions that commonly present with symptoms resembling ADHD."
-Methodology has been updated
-The abstract has been modified to include the identified conditions.
- The conclusion recommends a thorough medical evaluation before diagnosing ADHD and suggests involving physicians for medical clearance. The implications or potential impact of this recommendation has been added to reduce misdiagnosis rates and improve patient outcomes.
- Ambiguous language has been eliminated
Introduction
- Brief mention of the primary symptoms of ADHD had been added.
- While prevalence data is mentioned, it would be beneficial to briefly discuss the impact of these prevalence rates and why they vary. For instance, you could mention that the variation in prevalence is due to factors such as diagnostic criteria and geographical differences. (highlighted in the text)
- A transitional paragraph has been added
- The DSM-5 has been explained.
- the objective is now more explicitly stated.
- References had been added to enrich the background
h
Materials and Methods
The criteria used for selecting the literature and search strategies are specified
The section briefly mentions that "important findings are summarized below," but it doesn't provide any specific findings. Readers would benefit from a brief overview of key findings or themes identified during the review process.
Results
- Clear headings for each medical condition (e.g., "3.1. Seizure Disorder: Absence Seizure," "3.2. Diabetes Mellitus," etc.). had been added.
- the potential consequences or implications of such misdiagnoses of seizure is discussed
- Smooth flow among different medical conditions was not achieved due to the lack of connection between the different medical disorders.
Discussion and Conclusions
ü The suggestion to modify the DSM-6 criteria to exclude the diagnosis of ADHD when symptoms are attributable to medical conditions is a significant proposal. A more detailed rationale for this recommendation was added.
- Practical guidance on what the medical evaluation should entail is now provided.
-Example of how misdiagnosis can have life-threatening consequences has been provided for the seizure disorder.
ü
-The potential reversibility of cognitive deficits in various medical conditions was not discussed since there is no clear agreement in the current literature on this issue.
-Statement about the challenges of thorough medical evaluations has been added.
Please let me know of any other corrections I need to make.
Thanks again
Joseph Sadek

Reviewer 3 Report
Comments and Suggestions for Authors
The title of the paper appropriately conveys the key message of the review. Authors have presented this as a selective narrative review; however, it would be useful to conduct literature search in a defined way and should be described as such. Clarity on inclusion and exclusion criteria of articles would be helpful. The medical conditions that should be considered when a patient presents with behavioral symptoms suggestive of ADHD should also include iron deficiency state and anemia, inflammatory bowel disease, post-concussive syndrome, and unrecognized visual and hearing impairment. It would be useful to limit description of various medical conditions as they relate to symptoms that are similar to ADHD. Authors state that in the presence of medical conditions that explain ADHD type behavioral symptoms, a ADHD diagnosis should excluded; however, both, ADHD and a medical condition may be present in the same patient. Both are not mutually exclusive.
Comments on the Quality of English LanguageIn addition to comments to authors I would like to add that this paper needs some editing of English (including careful review for spelling errors). The message of the paper is an important one to consider in that often a patient present first to a mental health professional with ADHD symptoms, and a medical evaluation should be part of the overall evaluation.
Author Response
Dear Reviewer
I am extremely grateful for the time, dedication, and expertise that you put in this review. I agree with your comments. I added the inclusion and exclusion criteria of the search.
I also added section 3.6 that include iron deficiency state and anemia, inflammatory bowel disease, post-concussive syndrome.
I corrected the spelling errors as suggested.
Thanks again for the excellent review.
Joseph Sadek

Round 2
Reviewer 1 Report
Comments and Suggestions for Authors
I still think that it is possible include more critical information andn include proposed citations in the article.
Author Response
Dear Reviewer
Thanks again for the excellent suggestion. I have included the following paragraph in the discussion which I think would add an important dimension to this article"
ADHD diagnosis remains challenging and complex. It requires further research. Lauria’s neuropsychological theory explains another dimension of complexity to brain functioning particularly due to the complex representation of the cultural psychological process in the central nervous system. It was emphasized that neuropsychological assessment helps to understand complex functional changes in the human brain rather than localize a lesion."
The reference has been added to the list.
Thanks again
Joe